# Multilingual self-supervised speech representations improve the speech recognition of low-resource African languages with codeswitching

**Tolúlọpẹ́ Ògúnrẹ̀mí**     **Christopher D. Manning**     **Dan Jurafsky**

Stanford University
`tolulope@cs.stanford.edu`

## Abstract

While many speakers of low-resource languages regularly code-switch between their languages and other regional languages or English, datasets of codeswitched speech are too small to train bespoke acoustic models from scratch or do language model rescoring. Here we propose finetuning self-supervised speech representations such as wav2vec 2.0 XLSR to recognize code-switched data. We find that finetuning self-supervised multilingual representations and augmenting them with n-gram language models trained from transcripts reduces absolute word error rates by up to 20% compared to baselines of hybrid models trained from scratch on code-switched data. Our findings suggest that in circumstances with limited training data finetuning self-supervised representations is a better performing and viable solution.

## 1 Introduction

Over half of the world's population uses at least two languages regularly (Ansaldo et al., 2008). Despite this common occurrence, automatic speech recognition (ASR) models don't work well with speech that includes **code-switching**: when a speaker alternates between two or more languages or varieties within utterances (Myers-Scotton, 2017). For low-resource languages, we encounter two issues when attempting to address this problem: insufficient data for end-to-end-training and insufficient data for language modelling.

Recently, self-supervised pre-training of speech such as wav2vec 2.0 (Baevski et al., 2020) have proven to give very low error rates for English ASR. Although very costly to pre-train, the English models and cross-lingual (XLSR) representations (Conneau et al., 2020) are available for finetuning to efficiently make speech recognisers for many languages.

In this work we ask: *Does fine-tuning XLSR improve recognition of code-switched data over traditional training on code-switched data?* To test this phenomenon, we look at four African languages (isiZulu, isiXhosa, Sesotho, Setswana) code-switched with English. We also explore three questions about how to go about this fine-tuning process. We first experiment with different types of data to add to the codeswitched dataset in order to improve ASR performance, asking *1. Should we add monolingual data?* Many other methods incorporate language identification (language ID) into models, so we ask: *2. Does it help to add language identification in our pipeline (either explicitly or implictly)?* . We test this by augmenting utterances to implicitly identify the language and use a multi-task learning setup to learn frame-level language ID and ASR simultaneously. Finally, we ask: *3. Does a simple n-gram language model trained on the code-switched data improve performance despite the tiny amount of data?*. We use the codeswitched corpus to train bigram and trigram models which we use when decoding the models.

We find that finetuning multilingual pretrained models, augmented with a simple trigram language model, works well for recognizing code-switched data in low-resource languages, significantly better than prior methods of training bespoke models (CNN-TDNN-F acoustic model + LSTM language model) from scratch. We find that neither language ID nor adding monolingual data adds further performance gains and perhaps surprisingly, that adding monolingual data worsened model performance. Our findings suggest that in circumstances with limited training data, finetuning self-supervised representations are likely a better performing and viable solution.

## 2 Related Work

In speech processing, work on code-switching can be divided into code-switching detection (Rallabandi et al., 2018; Yılmaz et al., 2016; Wang et al., 2019) using language identification (Choud-

hury et al., 2017) and end-to-end recognition (Indra Winata et al., 2018). In this work, we look at both methods via finetuning of self-supervised representations, namely wav2vec 2.0 (Baevski et al., 2020). Language identification methods either identify the language before doing the ASR on the speech or have language ID trained in tandem with the acoustic model of representations. End-to-end recognition splits into two main approaches: a multilingual modelling with cross lingual representations (Li et al., 2019a; Luo et al., 2018; Zhang et al., 2022) and parallel modelling generating multiple transcriptions which are interpolated to result in one transcription with the highest likelihood (Ahmed and Tan, 2012; Lyu et al., 2006).

For low-resource languages, we encounter two issues when attempting to apply these methods: a lack of sufficient data for end-to-end training and a lack of sufficient data for neural language modelling in the low-resource language or the codeswitched language pair. The absence of a language model for the codeswitched pair leads to prior less computationally expensive methods to fail and the lack of sufficient data for the model to generalise, resulting in poor performance of models.

In our work, we focus on leveraging a pretrained self-supervised acoustic model, wav2vec 2.0 (Baevski et al., 2020) to finetune an existing multilingual acoustic model for our chosen language pairs. We incorporate language identification to see if this additional signal can improve performance given the small datasets.

## 3 Background

### 3.1 Languages

The languages used in this work are four South African languages and English. The South African languages are all Southern Bantu (SB) languages, in the Nguni and Sotho-Tswana branches. The English used in this work is English spoken with a South African accent.

### 3.2 Data

We use the South African corpus of multilingual code-switched soap opera speech (Niesler et al., 2018). It is a corpus of speech collected from 626 South African soap opera episodes, with utterances from four South African languages: isiZulu, isiXhosa, Sesotho and Setswana codeswitched with English.

| Language | No. speakers (millions) | Language Family |
|---|---|---|
| isiXhosa | 11.6 | SB: Nguni |
| isiZulu | 8.2 | SB: Nguni |
| Sesotho | 4.0 | SB: Sotho-Tswana |
| Tswana | 3.8 | SB: Sotho-Tswana |
| English | 380 | IE: Western Germanic |

Table 1: An overview of the languages used in this work. The South African languages are in the Nguni and Sotho-Tswana branches of the Southern Bantu (SB) language family and English is in the Western Germanic branch of the Indo-European (IE) language family.

For additional monolingual data in the languages, we use the isiZulu, isiXhosa, Sesotho, Setswana and English portions of the NCHLT Speech Corpus (Barnard et al., 2014) to add as monoingual supplementary finetuning data. We use the *NCHLT-clean* partition of the dataset. The datasets used in this work are summarised in Table 2.

| | Lang(s) | No. utts | Duration (hrs) |
|---|---|---|---|
| Soap Opera Corpus | Eng-Zul | 9347 | 5.45 |
| | Eng-Xho | 7941 | 3.14 |
| | Eng-Sot | 6303 | 2.86 |
| | Eng-Tsn | 6563 | 2.83 |
| NCHLT Corpus | isiZulu | 44673 | 56.2 |
| | isiXhosa | 46651 | 56.3 |
| | Sesotho | 57539 | 56.3 |
| | Setswana | 58414 | 56.3 |
| | English | 77412 | 56.4 |

Table 2: Summary of the data used in experiments from both the South African corpus of multilingual codeswitched soap opera speech (Soap Opera Corpus) and NCHLT-clean Speech Corpus (NCHLT Corpus).

### 3.3 Baseline Model

We compare our models to those trained from scratch on this data by Biswas et al. (2022). Their best performing acoustic model is a Kaldi-based (Povey et al., 2011) CNN-TDNN-F trained on all 5 languages and finetuned for each language pair. For language model decoding, the authors used a bidirectional LSTM architecture with a 256-dimensional embedding and 256-dimensional matrices. The LSTMs are trained on language pairs, resulting in four separate language models. We compare our methods to the best performing model for each language pair in this work.

## 4 Which additional data is helpful?

Given the low-resource natural of codeswitched speech datasets, we ask which type of data can best

supplement the codeswitched dataset to improve downstream results. To test this, we "pre-finetune" the model with additional data other than the Soap Opera Corpus data for each language pair, before finetuning it on the codeswitched language pair.

To test whether in-domain data is most useful, we pre-finetune the model with Soap Opera Corpus data from all four language pairs for 42000 steps. This model is then further finetuned with the Soap Opera Corpus data for each individual language pair alone for 12000 steps, resulting in the **+all 4 pairs** models.

To test whether adding monolingual data improves performance, we use NCHLT monolingual data from each language in a language pair, plus the data from the corresponding language pair in the Soap Opera Corpus data to pre-finetune models for 42000 steps. We then further finetune these models with Soap Opera Corpus data from that specific language pair, resulting in **+monolingual** models.

To compare the proposed methods with finetuning with solely Soap Opera Corpus data in the desired language pair, we finetune the model for 15000 steps with the Soap Opera Corpus data for that language pair, resulting in the **One pair** models.

Table 3 shows the results for these experiments with greedy decoding.

| Lang pair | Model type | WER |
|---|---|---|
| | One pair | 72.2 |
| xho-eng | **+all 4 pairs** | **59.0** |
| | +monolingual | 77.5 |
| | One pair | 60.8 |
| zul-eng | **+all 4 pairs** | **50.8** |
| | +monolingual | 67.6 |
| | One pair | 59.4 |
| sot-eng | **+all 4 pairs** | **50.2** |
| | +monolingual | 63.3 |
| | One pair | 51.4 |
| tsn-eng | **+all 4 pairs** | **42.7** |
| | +monolingual | 60.4 |

Table 3: Effects of additional data used in "pre-finetuning" on ASR performance. WER is word error rate of models. **+all 4 lang pairs** is "pre-finetuned" with in-domain codeswitched data from the Soap Opera Corpus and **+monolingual** is "pre-finetuned" with monolingual data in each language in the lamguage pair along with the Soap Opera Corpus data for that specific pair.

We see that across languages, using codeswitched-data from all four languages (i.e., "pre-finetuning" with Soap Opera Corpus data from all 4 languages) gives the best results on each South African language pair. The fact that adding data from three different languages helps on the 4th language is somewhat surprising, and points both to the importance of the similarity of the 4 languages, and to the fact that all data are from a single Soap Opera genre. By contrast, the genre difference from the monolingual read speech data is enough to severely hurt performance. In summary, when finetuning multilingual, self-supervised ASR models on low-resource codeswitched data, we find that matching domain and genre properties (such as the presence of codeswitching) is more important than adding monolingual data from the same language if the genre is a mismatch.

## 5 Does adding implicit or explicit language id information help?

Prior work has shown that for codeswitched ASR, simultaneously learning the language identification (language ID) and ASR improved the ASR performance (Luo et al., 2018; Li et al., 2019b; Zeng et al., 2019). Here we try to add language ID information in two ways: by augmenting the data and by training a classifier.

We experiment with augmenting the Soap Opera Corpus utterances to encapsulate the bilingualism in the utterances in lieu of explicit language labels or timestamps. For each language pair, we use two methods: language specific casing and language specific tags. For language specific casing, we double the vocabulary size by giving each language a specific case, e.g., English in uppercase and isiZulu in lowercase. We then finetune wav2vec 2.0 XLSR 300M with this data for 12000 steps resulting in **+casingID** models for each language pair. For language specific tags, we put opening and closing tags on either side of the text in a specific language. We then finetune wav2vec 2.0 XLSR 300M with this data for 12000 steps resulting in **+tagsID** models for each language pair.

Casing: **WHAT IF etholwa amaphoyisa kuqala**

Tags: ***<eng>* what if *</eng>* *<zul>* etholwa amaphoyisa kuqala *</zul>***

**Example 1**: Demonstration of implicit addition

of language information to our models through language-specific casing and language-specific tags.

To train a language ID classifier on our data, we add a frame-level classification head to the wav2vec 2.0 XLSR encoder. We use the timestamps in the corpus to label frames with either English or the South African language, and train a model with cross-entropy loss. The results of the language ID models are in Table 4.

| Language Pair | Lang ID Accuracy |
|---|---|
| English-isiZulu | 97% |
| English-isiXhosa | 98% |
| English-Sesotho | 96% |
| English-Setswana | 97% |

Table 4: Results from frame-level language identification of the four South African languages and English

The frame-level language ID models work well, so we try a multi-task setting in hopes of improving the model performance. We learn language ID and ASR at the same time, summing the weighted loss of the two tasks. The loss calculation is summarised in Equation 1. As ASR is the priority, we always keep the CTC weight higher than the language ID weight. The resulting models are the **+multitaskID** models, with each language pair finetuned for 12 00 steps. The model architecture is visualised in Figure 1 .

$$Loss_{CTC+LID} = \lambda_{CTC}L_{CTC} + (1 - \lambda_{CTC})L_{LID} \tag{1}$$

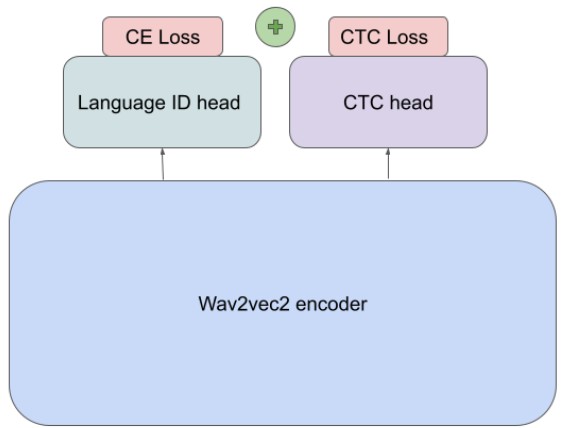

Figure 1: Our multi-task learning setup for combining frame-level language ID with CTC by a weighted sum of the losses.

| Lang pair | Model type | WER |
|---|---|---|
| xho-eng | One pair | 72.2 |
| | +tagsID | 83.4 |
| | +casingID | 87.9 |
| | +multitaskID | 75.2 |
| zul-eng | One pair | 60.8 |
| | +tagsID | 80.8 |
| | +casingID | 80.9 |
| | +multitaskID | 64.2 |
| sot-eng | One pair | 59.4 |
| | +tagsID | 76.3 |
| | +casingID | 89.4 |
| | +multitaskID | 65.6 |
| tsn-eng | One pair | 51.4 |
| | +tagsID | 72.6 |
| | +casingID | 86.6 |
| | +multitaskID | 64.5 |

Table 5: Effects of incorporating language ID on ASR performance. WER is word error rate of models. **+tagsID** uses language specific tags around utterances in the dataset and **+casingID** uses one case per language (e.g. uppercase for English and lowercase for isiZulu). Models trained to learn both language ID and ASR at the same time during finetuning are referred to as **+multitaskID** models. The **+multitaskID** models work better that **+tagsID** and **+casingID**. But none of the language ID models work as well as the baseline of not using Language ID at all (the "One pair" row).

The results of our experiments are in Table 5. For the multi-task setup, the results with the best language ID and CTC weights are reported.

The multi-task learning setup improves performance downstream over language specific casing and tags, but not over further fine-tuning, possibly due to the model being hindered rather than helped trying to learn two tasks at once.

Language specific casing does not improve model performance, it actually worsens the models compared to the baselines. This is likely due to the unnecessary doubling of the vocabulary.

Language ID tags work better than the casing across languages, however they do not outperform finetuning without tags. This is likely due to the fact that the tags do not correspond to any speech, so the introduction of them creates initial confusion.

In summary, adding language identification information does not improve ASR performance on our code-switched dataset. This could be due to the lack of data available for training, the fact

that the character sets for our 5 languages are all overlapping, or the fact that our experiments consist of finetuning and not end-to-end pretraining. Other work that uses multitask learning for code-switched speech recognition (Li et al., 2019b; Zeng et al., 2019; Song et al., 2022; Winata et al., 2018) has shown success with a language pair with an non-overlapping character set: English and Mandarin Chinese. Those English/Chinese models are also trained from scratch end-to-end, so it is possible that incorporation of language ID is more useful during training and less useful at later stages such as finetuning.

## 6 Does a language model improve performance?

For our experiments thusfar, we do greedy decoding from the wav2vec 2.0 model finetuned with a CTC head. Could adding language model information improve performance? The baseline system with which we are comparing used an LSTM language model, suggesting that this information might be useful.

In this section, we study whether using the transcripts from the Soap Opera Corpus as training data for a small n-gram language model could improve accuracy. We train separate bigram and trigram (word) language models using KenLM (Heafield, 2011) from each of the 4 language-pair datasets, and then use this language model in decoding.

The language model results for the best finetuned models per language pair are presented in Table 6.

| | xho-eng | zul-eng | sot-eng | tsn-eng |
|---|---|---|---|---|
| Baseline | 48.7 | 43.3 | 48.5 | 43.5 |
| Greedy | 59.0 | 50.8 | 50.2 | 42.7 |
| 2-gram | 26.7 | 25.5 | 30.6 | 28.9 |
| 3-gram | **22.1** | **22.3** | **23.4** | **21.7** |

Table 6: Effect of language modelling on ASR performance (measured in WER). The numbers in the baseline raw are taken from (Biswas et al., 2022); their system (which includes an LSTM language model) is compared to wave2vec 2.0 finetuned on the Soap Opera Corpus data, using greedy decoding (no LM) as well as bigram, and trigram n-gram models trained with the Soap Opera Corpus data. Without n-gram language models, the baseline model outperforms finetuning wav2vec 2.0. However, training an n-gram language model with the ASR data improves over the baseline.

Although greedy decoding does not work better than the baseline (CNN-TDNN-F acoustic model plus a bidirectional LSTM model) since the baseline has a language model, we find that the finetuned models equipped with a simple n-gram language model consistently beat baseline models. These results suggest that fine-tuning large pretrained models with only very simple language model support can be a better solution in low-resource scenarios.

## 7 Conclusion

In this work, we have finetuned wav2vec 2.0 XLSR with codeswitched data of South African languages and English. We found that this system augmented with a simple bigram or trigram language model beats baseline models trained with LSTM language models. We also found that it helps to add data from other languages, albeit very related languages and in the exact the same genre/domain.

We were not able to improve the model with various kinds of language ID information; these methods may see more success for languages with character sets that overlap less, or when there is enough data to train an end-to-end model from scratch.

This work demonstrates a method to train ASR models on codeswitching data with relatively minimal computation and a very basic n-gram language model, suggesting a direction for addressing an important task in the low-resource settings that characterise many of the world's languages.

## 8 Acknowledgements

We would like to thank the reviewers for their comments and suggestions. This research was funded in part by NSF award number IIS-2128145 and in part by a Stanford School of Engineering Fellowship to TO. CM is a fellow in the CIFAR Learning in Machines and Brains program.

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
