# OpenReview forum: "Multilingual self-supervised speech representations improve the speech recognition of low-resource African languages with codeswitching"
_EMNLP/2023/Workshop/CALCS — EMNLP 2023 Workshop CALCS_

### Official Review · Reviewer_Zacj · 2023-10-03
**The paper tackles the problem of transcription of African-English code-switch speech dataset. The authors propose to fine-tune wave2vec 2.0 XLSR model to recognize code-switched data. They explore multiple setups: using only codeswitched data, monolingual data,  finetuning with language specific  casing, language-specific tags, multitask settings (lang id and ASR). They showed that finetuning the finetuned models of 5 South African languages with codeswitched data performed the best.**

**Rating:** 3
**Confidence:** 4

**Review:**

Strength
- they tackled the problem of transcription of 5 low resource African-English code-switch speech dataset
- they explored multiple settings: using only codeswtiched data, monolingual data, finetuning with language specific  casing, language-specific tags, multitask settings (lang id and ASR)
- they showed the importance of codeswitched data for finetuning  to reduce the word-error rate

Weakness
-  the paper is not well written (typos: Speed -> speech; EngZul -> Eng-Zul; 15 000 -> 15,000)
- the authors mention that they use MSE loss to train a classification of langID.
- the different experimental settings are not properly explained
- the performance improvement seems to be coming mainly from addition of 2,3-gram language models

**Candidate For Best Paper:**

No

**Reason For Best Paper:**

N/A

**Related:**

5: It is very related to the workshop.

---

### Official Review · Reviewer_Fmjw · 2023-10-04
**The paper introduces a self-supervised approach  for speech recognition in low-resource code-mixed data. The paper stands out for its thorough exploration of the experiments. It excels in providing detailed analysis of the results, with accurate explanations .**

**Rating:** 4
**Confidence:** 5

**Review:**

## Reviewer Scores
- Appropriateness (1-5): 4
-  Clarity (1-5): 5
- Originality / Innovativeness (1-5): 4
- Soundness / Correctness (1-5): 4
- Meaningful Comparison (1-5): 3
- Thoroughness (1-5): 4
- Impact of Ideas or Results (1-5): 4


## Strong Points:
1. The paper demonstrates a comprehensive exploration of experiments.
2. It provides an in-depth analysis of the results, accompanied by precise explanations.
3. The paper is well-written

## Concerns and question to the authors:
1. In line 055, the word "with" is missing.
2. Related Work section can be expanded
3. The caption for Table 4 is incomplete.

**Candidate For Best Paper:**

Yes

**Reason For Best Paper:**

Insightful, Good exploration and impactful

**Related:**

5: It is very related to the workshop.